# An Evaluation of the Predatory Function of *Orius strigicollis* (Poppius) (Hemiptera: Anthocoridae) on *Megalurothrips usitatus* (Bagnall) (Thysanoptera: Thripidae)

**DOI:** 10.3390/insects16030236

**Published:** 2025-02-21

**Authors:** Zuying Fu, Yuanrun Cheng, Yifan Cui, Changyu Xiong, Ziyu Cao, Ying Wang, Rong Zhang, Chang Liu, Wei Sun, Liping Ban, Yao Tan, Shuhua Wei

**Affiliations:** 1Institute of Plant Protection, Ningxia Academy of Agricultural and Forestry Sciences, Yinchuan 750002, China; 19896202015@163.com (Z.F.); cyr20010216@163.com (Y.C.);; 2College of Horticulture and Plant Protection, Inner Mongolia Agricultural University, Hohhot 010010, China; 3College of Grassland Science and Technology, China Agricultural University, Beijing 100193, China; lipingban@cau.edu.cn; 4College of Biological Science & Engineering, North Minzu University, Yinchuan 750002, China

**Keywords:** natural enemy, predation function, intraspecific interference, predation preference, biological control

## Abstract

*Megalurothrips usitatus* (Bagnall) is one of the most important pests that harm legumes—namely alfalfa (*Medicago sativa* L.). The pest is harmful, has strong reproductive abilities, and can transmit plant viruses. In addition, not only does the long-term use of chemical pesticides make the pest resistant, but pesticide residues also have a negative impact on the ecological environment, meaning that control is difficult. *Orius strigicollis* (Poppius) is a common predatory natural enemy in the field which can prey on several kinds of pests. In our experiment, we found that the predation ability of *O. strigicollis* on *M. usitatus* strengthened with an increase in prey density, while its searching efficiency decreased with an increase in prey density. However, the intensity of natural enemy apportionment competition increased with an increase in their own density. These results provide a scientific basis for the use of *O. strigicollis* to control *M. sativa* pests and develop biological controls.

## 1. Introduction

Alfalfa (*Medicago sativa* L.) has an illustrious history, stretching back through the ages [1]. It is a high-quality perennial legume forage, which is not only nutritious but also has a high protein content and high crude fiber digestibility; therefore, it is known as the “king of forages” [2,3,4]. *M. sativa* can also play an important role in soil improvement because of its nitrogen fixation capacity [5]. In China, the planting area of *M. sativa* has been expanding with the ongoing reforms in the country’s crop industry, and with that expansion, the problem of pest control has become more prominent [6]. Investigations have shown that thrips (Thysanoptera: Thripidae) are the major pests of *M. sativa* in the northwestern regions of China [7], while *Megalurothrips usitatus* (Bagnall) is a major *M. sativa* pest infesting *M. sativa* in Ningxia [8].

*Megalurothrips usitatus* (Bagnall) is known as bean flower thrips, Asian bean thrips, blossom thrips, or flower thrips and is widely distributed in Asia [9,10]. *M. usitatus* undergoes six developmental stages on the host plant: egg, first- and second-instar larva, prepupa, pupa, and adult. The second-instar larval stage is the longest and most active [11]. Tang et al. [12] reported that the lifespan of an adult *M. usitatus* at 26 °C ranges from 10.63 to 18.11 days, and second-instar larvae and adult thrips cause the most damage to the legume, mainly by feeding on its flowers. These studies show that the second-instar larvae and adults of *M. usitatus* cause the most damage to legumes. However, when the flowers of the leguminous plant are no longer available, *M. usitatus* adapts and starts feeding on the young leaves and pods [12,13], not only deforming and corrupting the host plant but also transmitting Tobacco Streak Virus (TSV) [14,15]. It can cause an 80–100% yield loss in legume crops in severe cases, leading to complete crop failure and significant economic losses for farmers [16]. At present, chemical insecticides are the main method for controlling thrips [17,18]; however, the long-term use of chemical pesticides has a negative impact on the environment, humans, and other living organisms by making pests resistant and upsetting the biological balance between natural enemies and pests [19,20,21,22]. In addition, *M. usitatus* is difficult to control, because it has characteristics such as strong concealment, rapid reproduction, and a short generation cycle and can be hidden in closed spaces to avoid chemical sprays [19,23].

The release of predatory natural enemies does not affect the environment or destroy ecological diversity, and it is one of the most important measures for integrated pest management [24]. Of these, *Orius* spp. (Hemiptera: Anthocoridae) are the most important group of predators that prey on thrips [25], and *O. strigicollis* has been commonly used as a biocontrol agent against thrips by virtue of its high search ability and consumption rate [26,27]. Wang et al. [28] found that the release of *O. strigicollis* in the field reduced the population density of *M. usitatus*. Dai et al. [29] compared *O. minutus* (Linnaeus), *O. nagaii* (Yasunaga), *O. sauteri* (Poppius), and *O. strigicollis* found that the latter has a significant potential for controlling *M. usitatus*. Related studies have shown that *O. strigicollis* can effectively control *Thrips hawaiiensis* (Morgan) [30], *Dendrothrips minowai* (Priesner) [31], *Frankliniella occidentalis* (Pergande) [32], *Thrips palmi* (Karny) [26], and other pests.

Based on the above studies, our study was conducted under indoor constant-temperature conditions, using second-instar larvae and adults of *M. usitatus* as prey and *O. strigicollis* adults as their natural enemy. By constructing models of functional responses to predation and calculating intraspecific interferences and predation preferences, it is possible to more accurately predict and understand the complex interactions between predatory natural enemies and prey. This will not only help deepen our understanding of predatory natural enemies and pest biology but also facilitate the development of more effective tools for agricultural practices to reduce *M. usitatus* infestation on *M. sativa*.

## 2. Research Materials and Methodology

### 2.1. Materials and Methodology

Insects and plants for testing: *O. strigicollis and M. usitatus* were collected from a *M. sativa* planting base in Xixia District, Yinchuan City, Ningxia Hui Autonomous Region (N38°38′59′, E106°9′6′). *M. usitatus* were collected and then reared in an artificial climatic chamber with a photoperiod of 14L:10D in succession with cowpeas (*Vigna unguiculata*). *O. strigicollis* was fed with *M. usitatus* in a plastic container in the same artificial climatic chamber. After *M. usitatus* had laid eggs and hatched, 2-day-old second-instar larvae and adults were selected as prey, and healthy and active adults of *O. strigicollis* of unknown ages were selected as the natural enemies for experiments. The individuals were starved for 24 h before the experiment [33].

### 2.2. The Predatory Function and Feeding Ability of O. strigicollis on Second-Instar Larvae and Adults of M. usitatus

*O. strigicollis* were transferred individually to plastic tubes (d = 9 cm; h = 11 cm). Then, second-instar larvae and adults of *M. usitatus* were placed into each test tube at different densities (10, 20, 30, 40, and 50 prey per predator, respectively). After 24 h, the number of thrips that had been eaten by *O. strigicollis* in each of the tubes was counted. Each density included 4 replicates.

### 2.3. Intraspecific Interference Experiment Determining O. strigicollis’s Predatory Activities on Adults of M. usitatus

The intraspecific interference response is a natural enemy’s own density interference response. *O. strigicollis* were placed in tubes at densities of 1, 2, 3, 4, and 5 individuals per tube, and the density of *M. usitatus* adults was set as 100 individuals per tube. The tubes were then placed in an artificial climatic chamber, with the same feeding conditions as described in Section 2.1. After 24 h, the number of *M. usitatus* adults that had been eaten by *O. strigicollis* was counted. Each treatment included 4 replicates.

### 2.4. Predation Preference of the O. strigicollis on Second-Instar Larvae and Adults of M. usitatus

Predation selection tests were conducted by selecting second-instar larvae and adults of *M. usitatus*. The second-instar of *M. usitatus* and adults were placed in plastic tubes (d = 9 cm; h = 11 cm) in ratios of 10:10, 20:20, and 30:30, respectively, and then, one *O. strigicollis* was placed in the plastic tubes. After 24 h, the number of *M. usitatus* larvae and adults that had been eaten by *O. strigicollis* was counted. Each treatment included 3 replicates.

### 2.5. Data Analysis

The predation function responses were fitted according to the Holling II disk equation (Holling 1959) [34]: *N_a_* = *aTN*/(1 + *aT_h_N*). The formula for the searching efficiency of *O. strigicollis* on *M. usitatus*, *S* = *a*/(1 + *aT_h_N*), was fitted [33,34,35]. In the formulas, *N* is the density of prey, *N_a_* is the number of predators at the corresponding density, *a* is the instantaneous attack rate, *T* is the total time of the experiment (1d in this experiment), and *T_h_* is the predator handling time for 1 prey (= T divided by the maximum predation rate).

For the intraspecific interference experiment with the *O. strigicollis* and *M. usitatus* adults, the Hassell [36] model equation was used for the fitting of the calculations to assess the disturbance effects. The formula was as follows: *E* = *QP*^−*m*^ = *N_a_*/*NP*. Here, *E* is the search rate, *Q* is the quest constant, *p* is the density of predators, and *m* is the mutual interference constant. The method of Zou et al. [37] was used to calculate the intensity of apportioned competition, with *I* = (*E*_1_ − *E_P_*)/*E*_1_, where *E*_1_ is the predation rate for 1 predator, and *E_P_* is the predation rate for the predators with a density of *p* [37].

The predation preference of the *O. strigicollis* on 2nd-instar larvae and adults of *M. usitatus* was evaluated using the evaluation method of Zhou et al. [38]: *C_i_* = (*Q_i_* − *F_i_*)*/*(*Q_i_* + *F_i_*), *F_i_* = *N_i_*/∑*N_i_*, and *Q_i_* = *Na_i_*/∑*Na*. Here, *Q_i_* is the predator’s proportion of prey for the *i*th prey species, *F_i_* denotes the proportion of the *i*th prey species among all the prey species, *N_i_* is the number of the *i*th prey species in the environment, *Na_i_* is the number of the *i*th prey species individuals that are preyed upon by the predator, and *Na* is the total amount of prey taken at the corresponding density. When *C_i_* = 0, it means that the predator has no preference for the *i*th prey; when 0 < *C_i_* < 1, it means that the predator has a positive preference for the *i*th prey; and when −1 < *C*_i_ < 0, it means that the predator has a negative preference for the *i*th prey [38].

The experimental data were collated using Office 2021, and SPSS 26.0 was used for statistical analysis of the data and one-way ANOVA. Duncan’s New Complex Polar Deviation method was applied for the significance of difference test, and GraphPad Prism 9.5.0 software was used for graphing.

## 3. Results

### 3.1. Functional Responses and Search Efficiency During Predation by O. strigicollis on Second-Instar Larvae and Adults of M. usitatus

The predation behavior of *O. strigicollis* on *M. usitatus* is to suck fluids from the thrips’ body by piercing the head (Figure 1D), thorax (Figure 1A,E), and abdomen (Figure 1B). The bodies of the *M. usitatus* larvae and adult thrips appeared to be emptied (Figure 1C,F).

The predation function response model of *O. strigicollis* on second-instar larvae and adults of *M. usitatus* was fitted according to the Holling type II disk equation, with *Na =* 1.022*N*/(1 + 0.013*N*) and *Na* = 1.003*N*/(1 + 0.019*N*), respectively. The instantaneous attack rate of *O. strigicollis* adults on second-instar larva thrips of *M. usitatus* was 1.022, the handling time was 0.013, the predation capacity was 78.62, and the maximum daily predation could reach 76.92. *O. strigicollis* had an instantaneous attack rate of 1.003 against adults of *M. usitatus*, while the handling time was 0.019, the predation capacity was 52.79, and the maximum daily predation was 52.62. The correlation coefficients of the resulting equations were 0.929 and 0.921, respectively (Table 1), indicating that predation on *M. usitatus* by *O. strigicollis* was significantly correlated with the prey density. The single daily consumption and search efficiency of *O. strigicollis* on second-instar larvae and adults of *M. usitatus* are shown in Figure 2. The daily consumption of *O. strigicollis* adults on *M. usitatus* increased with prey density with a positive relationship (Figure 2A), and the search efficiency decreased with the prey density with an inverse relationship (Figure 2B). The daily predation consumption and search efficiency on second-instar larvae of *M. usitatus* were significantly higher than those on adults of *M. usitatus* (*p* < 0.05).

### 3.2. Intraspecific Interference Experiment with O. strigicollis and Adults of M. usitatus

The densities of *O. strigicollis* were 1, 2, 3, 4 and 5 individuals per tube, and the density of *M. usitatus* adults was 100 individuals per tube. The predation amount, predation rate, and intensity of predation competition of *O. strigicollis* on *M. usitatus* were calculated for the different densities (Table 2). With a constant density and space of *M. usitatus*, the consumption and predation rates of *O. strigicollis* on *M. usitatus* decreased with their own density, and the intensity of apportioned competition increased with their own density, indicating mutual interference and competition among *O. strigicollis* individuals. The disturbance response equation for *O. strigicollis* predation on adults of *M. usitatus* was fitted with *E = QP^−m^* as *E* = 0.394*P*^−0.731^, with a correlation coefficient of 0.998, which can be used to describe the disturbance of *O. strigicollis’s* own density. Fitting the relationship between the intensity of apportioned competition (*I*) from *O. strigicollis* predation on *M. usitatus* and the logarithm of their own density (lg*P*), the resulting model was *I* = 1.003lg*P* + 0.037, with a correlation coefficient of 0.981, indicating that the intensity of apportioned competition was significantly correlated with the density of the predators.

### 3.3. Predation Preference of O. strigicollis on Second-Instar Larvae and Adults of M. usitatus

The predation preferences of *O. strigicollis* on second-instar larvae and adults of *M. usitatus* are shown in Figure 3. *O. strigicollis*’s predation on second-instar larvae of *M. usitatus* was significantly greater than on adults in all cases (*p* < 0.05) (Figure 3A). When both the second-instar larvae and adults of *M. usitatus* were 10, 20, and 30 in number, *O. strigicollis* showed a preference index of C_i_ > 0 for second-instar larvae and −1 < Ci < 0 for *M. usitatus* adults (Figure 3B). This indicated that *O. strigicollis* showed a positive preference for second-instar larvae of *M. usitatus* and a negative preference for *M. usitatus* adults when the second-instar larvae and adults of *M. usitatus* coexisted in the same numbers.

## 4. Discussion

The functional response is a pivotal characteristic within the intricate relationships between predator and prey and an essential component of predator–prey models. It unveils how predatory insects manage pest populations via their predatory behaviors, offering essential indicators for assessing the efficacy of biological control mechanisms. By monitoring and analyzing the ability of natural enemies to feed on pests under different environmental conditions, it is possible to accurately determine their actual role and potential impact in nature [39]. In this experiment, we recorded the predation ability and search efficiency of *O. strigicollis* on *M. usitatus* second-instar larvae and adults, as well as the intraspecific disturbance response of *O. strigicollis*. These observations not only reveal the behavioral adaptability of predators in hunting behavior, more importantly, they demonstrate how predators adapt to complex ecosystems by adjusting their own behavior. The search patterns of insects can respond to external and internal environmental factors, and have a high degree of plasticity, which is of great significance for understanding ecological balance and biodiversity. Through further research, we can gain a deeper understanding of the predation mechanisms of natural enemies and their roles in nature [40]. Additionally, the dried-out appearance of the *M. usitatus* bodies post-predation serves as a visual indication of the effectiveness of *O. strigicollis*’s feeding mechanism, highlighting the precision and efficiency of its predatory behavior. This is consistent with the predatory behavior of *Orius sauter* on *Dendrothrips minowai* [41].

The Holling II functional is considered the most common predatory functional response class in arthropods, characterized by a hyperbolics curve. Starting from low prey density on the abscissa, the predation rate increases almost linearly. It is generally accepted for use in evaluation systems for biological insect control and has strong applicability [39,42]. In our experiments, the predatory function response indicated that the predation ability and maximum predation amount of *O. strigicollis* were higher on second-instar larvae than on adults of *M. usitatus*. The predation of natural enemies increased with the prey density, while the search efficiency decreased with the prey density. These results were consistent with the predation function response that we calculated using Holling type II and similar to the predatory functional response of *O. sauteri* and *O. tantillus* on *M. usitatus* [33,43]. This shows that *O. strigicollis* has the potential to rapidly control *M. usitatus*, in addition, it exhibits greater effectiveness in managing pests in their young stages. Therefore, comprehensive pest management targeting different stages of life is crucial [44].

When the density of *O. strigicollis* itself increases, the intensity of competition becomes more significant, which means that *O. strigicollis* needs to compete with other individuals for limited resources. Thus, there are intensifying interactions and competitive relationships among individuals. The inter-individual interaction significantly increases the level of conflict among natural enemies, which in turn reduces the efficiency of predation [45]. But to a certain extent, it enhances the adaptive ability of individuals in the population to adverse environmental conditions, and *O. strigicollis* can adapt to environmental changes by regulating its own density in order to maintain the survival and reproduction of its population. This process also provides us with an important perspective to understand the behavioral ecology of insects, whose biodiversity is not always static and unchanging, but can be affected by a variety of factors such as environmental factors and the density of individuals [46,47].

The predation preference results showed that *O. strigicollis* had a positive preference for *M. usitatus* second-instar larvae and a negative preference for *M. usitatus* adults when the second-instar larvae and adults coexisted at the same density. This was similar to the results regarding the preference of *O. strigicollis* for *F. occidentalis* larvae and adults and indicates the complexity of the interaction between natural enemies and prey at different growth stages [48]. With age, prey grow in size and defense compared to their larval stage, making it more difficult for natural enemies to feed on them, which means that natural enemies have to spend more time capturing and feeding on the prey. Insects do not “keenly sense” environmental changes in the same way higher animals do. Therefore, predators have developed a complex set of adaptive mechanisms over a long period of co-evolution [49]. Physiological differences in the different age stages of *M. usitatus* may provide different nutritional values for *O. strigicollis* in order to obtain the maximum energy, and the ease of predation on larvae enables *O. strigicollis* to conserve its stamina, which in turn affects its predation strategy [50,51].

In conclusion, studying the functional response of *O. strigicollis* to predation by *M. usitatus* will not only deepen our understanding of the relationship between *M. usitatus* and its natural enemies but will also provide important data support for pest management. Understanding the inter-relationships among these organisms is essential for the development of effective pest management strategies while preserving the ecological balance [52] so that appropriate measures can be taken to protect natural enemies, control pests, and protect plants from insect pests.

## Figures and Tables

**Figure 1 insects-16-00236-f001:**
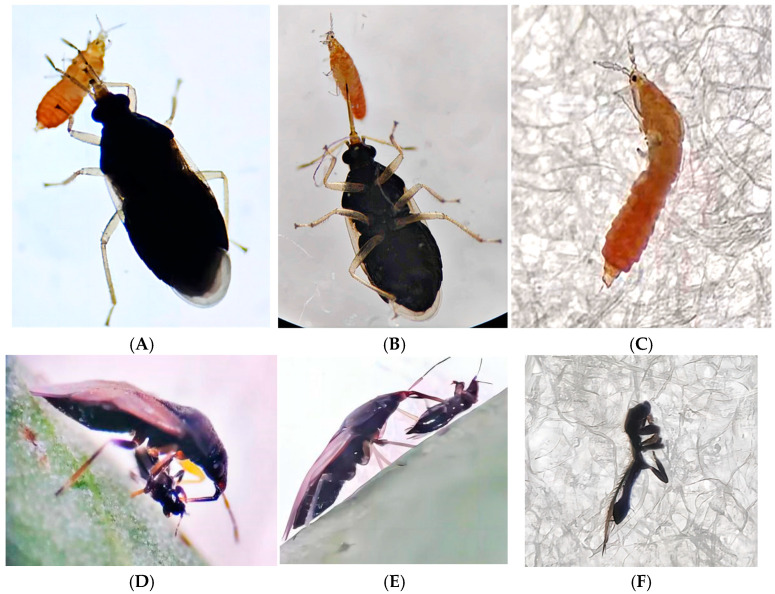
Feeding process of *O. strigicollis* adults on second-instar larvae and adults of *M. usitatus*. Note: Figures (**A**,**B**): feeding on second-instar larvae of *M. usitatus;* (**C**): larva’s condition after predation; (**D**,**E**): feeding on adults; (**F**): adult’s condition after predation.

**Figure 2 insects-16-00236-f002:**
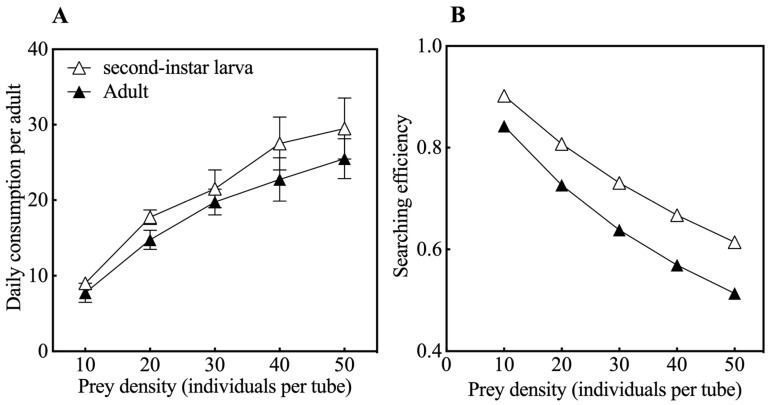
Daily consumption (**A**) and searching efficiency (**B**) of *O. strigicollis* adults on *M. usitatus*.

**Figure 3 insects-16-00236-f003:**
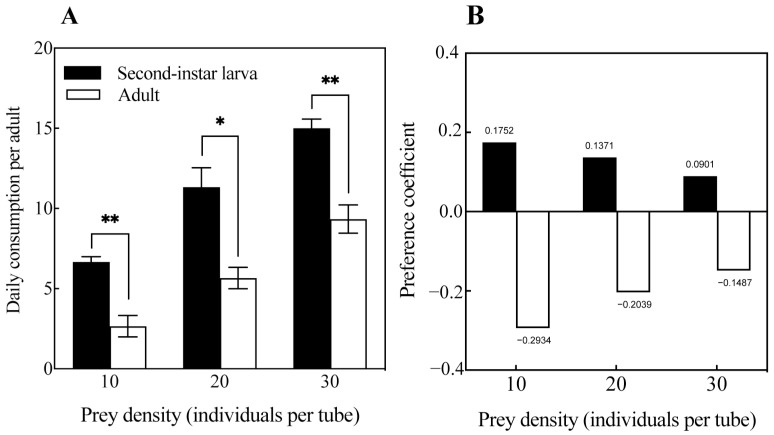
Daily consumption (**A**) and predation coefficient (**B**) of *O. strigicollis* on second-instar larvae and adults of *M. usitatus*. Note: * = *p* < 0.05, ** = *p* < 0.001.

**Table 1 insects-16-00236-t001:** Functional response equations and parameters for predation of *O. strigicollis* on *M. usitatus* of different ages.

*M. usitatus*’ Age	Predation Functional Response Equation	R^2^	Instantaneous Attack Rate (a)	Th Handling Time (d)	Predation Capacity (a/Th)	Daily Maximum Prey Consumed (1/Th)
Second-instar larva	*Na* = 1.022*N*/(1 + 0.013*N*)	0.929	1.022	0.013	78.62	76.92
Adult	*Na* = 1.003*N*/(1 + 0.019*N*)	0.921	1.003	0.019	52.79	52.62

**Table 2 insects-16-00236-t002:** Predation rate and intensity of apportioned competition of *O. strigicollis* adults against *M. usitatus* adults.

Density of Predators (Individuals per Tube) (*p*)	Predation Capacity (Number of Preys/Predators) (*Na*)	Predation Rate (*E*)	Intensity of Apportioned Competition (*I*)	F	df	*p*
1	39.25 ± 2.17 b	0.393	0.000	12.896	19	0.017
2	23.75 ± 2.90 c	0.238	0.395			
3	18.42 ± 1.25 a	0.184	0.531			
4	14.13 ± 1.84 a	0.141	0.640			
5	11.60 ± 2.42 a	0.116	0.704			

Note: Data are mean ± SE. Different lowercase letters in same column indicate significant difference, determined by means of Duncan’s new multiple range test (*p* < 0.05).

## Data Availability

The data presented in this study are available on request from the corresponding author.

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
