# Peer review of "An Evaluation of the Predatory Function of Orius strigicollis (Poppius) (Hemiptera: Anthocoridae) on Megalurothrips usitatus (Bagnall) (Thysanoptera: Thripidae)"

_insects, 2025, doi:10.3390/insects16030236_

Round 1
Reviewer 1 Report
Comments and Suggestions for Authors
The authors investigated the relationship between Orius strigicollis and Megalurothrips usitatus in laboratory conditions. They studied predatory bugs feeding on larvae and adults of M. usitatus in several experimental combinations. Such studies are essential in integrated pest management. To understand the relationships between predator and prey, the authors used several equations describing predator responses, predation rates, and the efficiency of O. strigicollis in finding M. usitatus. My comments on this manuscript are listed in the attached file in the Authors' Section.
The main remarks are:
1. In the entire text - English is insufficient and very poor
2. Too long sentences make the text unclear.
3. The rules for referencing literature in the text are inconsistent with those in the MDPI journal: lack of brackets, commas, or dashes.
4. The References are not prepared with the rules of MDPI journals - this chapter needs correction.
5. Authors should read the entire manuscript carefully to check for spelling errors and editorial rules.

The quality of English is relatively poor and needs severe correction.
Reviewer 2 Report
Comments and Suggestions for Authors
Thrips constitute an important group of pests and research into the biological possibilities of their control is very much needed. The authors investigated the predatory abilities of Orius strigicollis towards Megalurothrips usitatus in laboratory conditions. The paper contains many interesting results but requires major improvements before publication.
Many (perhaps most) sentences need revision due to grammatical issues, so the paper needs to be carefully reviewed by a person with a thorough knowledge of English before it can be published. There are many changes needed in general in the manuscript. Results reported in tables and figures and the text should be avoided. There is a lack of clarity in the calculations and some statistical analyses. The discussion needs to be expanded and improved. It should not repeat the results. The numbers of cited references should be in brackets, in many places spaces and dashes are missing.
Other Comments
Simple Summary
L 14 and L 18 Add the name of the first author of the scientific description of the species Megalurothrips usitatus (Bagnall) and Orius strigicollis (Poppius).
Abstract
L 25-26 Add the name of the first author of the scientific description of the species Megalurothrips usitatus (Bagnall) and Orius strigicollis (Poppius).
L 29 Change “2nd instar nymphs” to 2nd instar larvae, also in the rest of the text of manuscript
Keywords:
Do not repeat words contained in the title: Megalurothrips usitatus; Orius strigicollis;
Introduction
L 41 put the citation of article 1 in brackets, this applies to all cited references
L.48-52 Divide this long sentence into two shorter ones because it is not understandable and add publication number 5.
It is a high-quality perennial legume forage that is nutritious with high protein content and high crude fiber digestibility, so it is known as the "king of forages [2-4]. Also, alfalfa can play an important role in soil improvement because of nitrogen fixation capacity [5].
L 55. Change “The investigations have showed” on have shown
L 58. “Megalurothrips usitatus is an important pest of legume crops ..” this part of the statement is included in the previous sentence and can be omitted.
L 60 adds a dash between 9 and 10 [9-10]. Similarly in other lines e.g. 67, 68…
L 60-62 Should be “first and second instar larvae, prepupa, pupa and adults, and the second instar have the longest development and is very active”.
L62-64. Change ‘The life span of M. usitatus adult at temperature of 26 degrees Celsius 62 was 10.63 days at the shortest and 18.11 days at the longest12.So these researchs show that 63 the 2nd instar nymph and adult of M. usitatus cause the most damage to legume” to “Tang et al. [12] report that the lifespan of an adult M. usitatus at 26oC ranged from 10.63 to 18.11 days and 2nd instar larvae and adult thrips cause the most damage to the legume by feeding mainly on flowers’
L 64 Delete sentence: “Both the 64 larvae and adults of M. usitatus consume flower buds’
L 68 Change “It can causes 80-100 percent to “It can cause 80-100%..
L 91 Change “2nd instar nymph” to “2nd instar larvae”
L 92 Explain what you mean by “predatory function”, it was the feeding efficiency, the efficiency of searching for prey?
L92-95. This sentence is not understandable. Edit it so that it is clear which specific models you mean.
Research Content and Methodology
The subchapter titles in the Research Content and Methodology chapter do not reflect conducted experiments. Each subchapter should explain what indicators were obtained as a result of the tests performed.
L 108 How old were the adult thrips?
L 108-109 Add “of unknown-age” Healthy and active unknown-age adults of O. strigicollis were selected …
L 108. In what conditions and for how long was O. strigicollis kept after being collected from the field?
L 110 What do Functional Responses mean? Since the number of thrips consumed by each O. strigicollis was studied, perhaps the title should be different. Maybe “The feeding ability of O. strigicollis on 2nd Instar larvae and adult of M. usitatus” will be better.
L 111 Change “Instar Nymph” to “Insar Larvae”
L 115. In total, for each number of thrips, the experiment was performed only 4 times. (4 plastic tubes). This number of repetitions seems too small in this type of experiment. How long the experiment lasted?
L.110 What did you mean by Intraspecific Interference? The sentence is not clear.
L 114-115. “After 24h, the number of M. usitatus adults prey consumed by each O.strigicollis was recorded.” I think the number of thrips eaten by O. strigicollis in each of the tubes was calculated.
L 123 Change “2nd instar nymph” to “2nd instar larvae”
L 129. Also, three repetitions seem too small in this type of experiment. How long the experiment lasted?
L 136 How did you know the values of a and maximum predation rate?
L 144 How did you know the values of E1 and EP?
Results
L158 Change “2nd instar nymph” to “2nd instar larvae”
L160. Start the sentence with a capital letter
L 183-184 Change “2nd instar nymph” to “2nd instar larvae”.
L 164-172 Do not repeat text in the table and in the text (table 1)
L 170 “…the handing time was 167 0.013d….” remove d
L 191-195 Do not repeat text in the table and in the text (table 2)
L 208 Enter the “df’ value at the end of the sentence, also in L 219
Table 2 Enter the value of F and p in the table
L 209 Change column to row
L213-214 It would be worth adding that no significant differences were found….
Discussion
L 226 remove (Anno Domini)
L 224-242 The first paragraph of the discussion belongs to the introduction more than to the discussion…
L 243-248 This part of the discussion belongs to the results more than to the discussion…
L 277-296 The conclusions should be improved
Comments on the Quality of English Language
Many (perhaps most) sentences need revision due to grammatical issues, so the paper needs to be carefully reviewed by a person with a thorough knowledge of English before it can be published.
Reviewer 3 Report
Comments and Suggestions for Authors
Evaluation of the Predatory Function of the Orius strigicollis (Hemiptera: Anthocoridae) on Megalurothrips usitatus (Thysanoptera:Thripidae) by Zuying FU et al
This is a paper in which the authors evaluate the function of the predator O. strigicollis on immature and adult thrips Musitatus under laboratory conditions (indoor). The work presents some interesting data such as the density and intensity of predation, the preference for predation on immatures and adults of M. usitatus. However, even though the bioessays are well done, the writing presents little content for an extensive work, If we remove tables 1 and 2 because the information contained in them is already in the text, only figure 1 and table 3 remain. I therefore suggest (I don't know if this is within my remit) that the writing be converted into a scientific note or its equivalent in the journal. On the other hand, 11 authors seems excessive to me for the type of work they are proposing to publish. Below I make some comments and suggestions hoping they will help improve the writing..
Abstract
L42-43 = In conclusion, O. strigicollis …pest management of M. usitatus. This conclusion is very hasty since the work evaluated only a few parameters for biological control and in laboratory conditions. I suggest modifying it.
1. Introduction
L52-54 = Reference 5 is missing
L139-143 = Reference 41 is missing (it seems to be Zou et al.)
L81 = Orius spp. As far as I know spp. is not in italics, unless it is a requirement of the journal.
2.1. Materials and Methodology
L103-109 = Specify how the authors maintained both prey and predator breeding within the breeding chambers (plastic tubes mentioned above, cages etc.).
L103-129 = In none of the points of M and M (2.1 to 2.4) do the authors mention the age in hours of the adults of O. strigicollis, this becomes relevant since a young adult will not prey on the same amount of prey as an old adult. It also becomes important in order to replicate the bioassays. The authors also do not mention the age of the adult prey used in the bioassays..
3. Results
L160 = change the for The
L160-163 = the predation … emp-tied (Figure. 1-C, Figure .1-F). This knowledge is already very general and does not contribute anything new. I suggest deleting the paragraph together with figure 1.
L172 = … (Table 1)… I suggest removing this table as the results are already described in the text.
L186-187 = Delete table
L173-174 = … The single daily … are shown in Fig. 2. I suggest putting the letters that separate the treatment means in figures 2a and 2b.
L207-208 = I suggest deleting Table 2 as well. Same comment for Table 1, since the data is already described in the text.
L238-241 = As I mentioned in L160-163, this knowledge is very basic and does not contribute anything new.
L244-252 = There is no discussion as such in this text, the authors merely repeat the results. I suggest modifying it.
L277 = Put in italics O. strigicollis
Round 2
Reviewer 1 Report
Comments and Suggestions for Authors
Dear Authors,
Most of my comments included in the first review were corrected. You need to read the manuscript exactly and correct the citation of references according to the journal's rules. I have included all my comments in the margin of this attached manuscript.
I want to explain that the below-cited article contains a description of the development of thrips with the names of stages; you may cite it or not. Moritz, G. Structure, growth and development. In: Thrips as Crop Pests; Lewis, T., Ed.; CAB International, University Press: Cambridge, UK, 1997; pp. 15–63.

Author Response
Thank you for your positive feedback and professional modifications, we have made revisions to the manuscript based on your suggestions, please see the attachment.

Reviewer 2 Report
Comments and Suggestions for Authors
The authors responded to my comments and made corrections in most cases. Unfortunately, the discussion was not deepened.
There are still many errors in the work, especially editorial ones.
Please check all citations and include the number of each item in the text in brackets, e.g. [11], [12-13].
In the title, please add (Bagnall) after Megalurothrips usitatus (Bagnall) and (Poppius) after Orius strigicollis
In L 15 enter L. after Medikago sativa
In L 26 after alpha adds the full Latin name Medikago sativa L.
In the article only when mentioning the species of thrips and hemipteran for the first time give the full names, i.e. Orius strigicollis (Poppius) and Megalurothrips usitatus (Bagnall) and in the case of alpha alpha after the English name enter Latin name Medikago sativa L.
In the rest of the article use only M. usitatus, O.strigicollis, and M. sativa.
Comments on the Quality of English LanguageThe language has been improved, but I can't say if it's completely correct, I'm not a native speake
Author Response
The authors responded to my comments and made corrections in most cases. Unfortunately, the discussion was not deepened.
Reply: Thank you for your in-depth guidance. Based on your suggestions, we have made adjustments to the discussion.
There are still many errors in the work, especially editorial ones.
Please check all citations and include the number of each item in the text in brackets, e.g. [11], [12-13].
Reply: Thank you for your careful guidance. We had checked all citations and included the number of each item in the text in brackets.
In the title, please add (Bagnall) after Megalurothrips usitatus (Bagnall) and (Poppius) after Orius strigicollis
Reply: Thank you for your good suggestions. We had added (Bagnall) after Megalurothrips usitatus and (Poppius) after Orius strigicollis.
In L 15 enter L. after Medikago sativa
Reply: Thank you for your careful guidance. We had entered L. after Medikago sativa in the vision L15.
In L 26 after alpha adds the full Latin name Medikago sativa L.
Reply: Thank you for your excellent suggestions. We had added the full Latin name Medikago sativa L. it in the revision L26.
In the article only when mentioning the species of thrips and hemipteran for the first time give the full names, i.e. Orius strigicollis (Poppius) and Megalurothrips usitatus (Bagnall) and in the case of alpha alpha after the English name enter Latin name Medikago sativa L. In the rest of the article use only M. usitatus, O.strigicollis, and M. sativa.
Reply: Thank you for your careful guidance. We had modified these according to your suggestion in the revision.Please see the attachment

Reviewer 3 Report
Comments and Suggestions for Authors
Dear Kind regards,
Ms. Ivana Vostic
Assistant Editor
Insects
Regarding the revised version of the paper entitled "Evaluation of the predatory function of the Orius strigicollis (Hemiptera: Anthocoridae) on Megalurothrips usitatus (Thysanoptera:Thripidae)" by Zuying et al 2025, I can see that they almost carried out all my suggestions and requests (except removing tables 1 and 2, although these were improved, as well as figure 1). In addition, they did not remove the paragraph on lines 160-163. However, I can see that they made other improvements to the manuscript (perhaps suggested by other reviewers). In this version I can also see that there are some inconsistencies in the way of citing, in some cases the authors put the citations in parentheses and in others without them and without separating them with a hyphen. Therefore, I consider that the manuscript can now be accepted for publication in Insects after incorporating the minor revisions.
Author Response
Thank you for your professional guidance. Our manuscript has made revisions to the references cited. Please see the attachment.
